Evolution of heritable behavioural differences in a model of social division of labour

Vásárhelyi Zsóka 1 zsokavasarhelyi@gmail.com
Meszéna Géza 2
Scheuring István 3
1 Department of Plant Systematics, Ecology and Theoretical Biology, Eötvös Loránd University , Budapest , Hungary
2 Department of Biological Physics, Eötvös Loránd University , Budapest , Hungary
3 MTA-ELTE Theoretical Biology and Evolutionary Ecology Research Group, Eötvös Loránd University and the Hungarian Academy of Sciences , Budapest , Hungary
Perez-Acle Tomas
Electronic publication date: 2015 May 26
Publication date: 2015
Volume: 3
Electronic Location ID: e977
Received 2015 Mar 23; Accepted 2015 May 6
Copyright: © 2015 Vásárhelyi et al.
Copyright year: 2015
Copyright holder: Vásárhelyi et al.
License: This is an open access article distributed under the terms of the Creative Commons Attribution License, which permits unrestricted use, distribution, reproduction and adaptation in any medium and for any purpose provided that it is properly attributed. For attribution, the original author(s), title, publication source (PeerJ) and either DOI or URL of the article must be cited.
License URL: https://creativecommons.org/licenses/by/4.0/

Keywords: Division of labour, Behavioural syndrome, Personality, Adaptive dynamics, Cooperation, Specialization

Funding: Hungarian Scientific Research Fund N100299 K81628 This research was supported by the Hungarian Scientific Research Fund (OTKA) No: N100299 and K81628. The funders had no role in study design, data collection and analysis, decision to publish, or preparation of the manuscript.

==============================
The spectacular diversity of personality and behaviour of animals and humans has evoked many hypotheses intended to explain its developmental and evolutionary background. Although the list of the possible contributing mechanisms seems long, we propose that an underemphasised explanation is the division of labour creating negative frequency dependent selection. We use analytical and numerical models of social division of labour to show how selection can create consistent and heritable behavioural differences in a population, where randomly sampled individuals solve a collective task together. We assume that the collective task needs collaboration of individuals performing one of the two possible subtasks. The total benefit of the group is highest when the ratio of different subtasks is closest to 1. The probability of choosing one of the two costly subtasks and the costs assigned to them are under selection. By using adaptive dynamics we show that if a trade-off between the costs of the subtasks is strong enough, then evolution leads to coexistence of specialized individuals performing one of the subtasks with high probability and low cost. Our analytical results were verified and extended by numerical simulations.

Introduction

Behavioural differences among conspecifics of the same population have been reported from all kinds of living organisms, from bacteria to vertebrates, including humans (Dall, Houston & McNamara, 2004; Dingemanse et al., 2004; Stampsa, Briffa & Biro, 2012; Grinsted et al., 2013; Cordero & Polz, 2014). A bacterium’s behaviour is determined by its possession or lack of certain genes. However, as organismal complexity rises, behaviour also becomes more subtle and less predictable, reaching the greatest complexity in humans. Interestingly, while variation in bacterial behaviour is widely studied as a classical model of fluctuating or negative frequency dependent selection (Cordero & Polz, 2014), the adaptive value of individual variation in animal personality, especially in humans, is still the subject of debate (Penke, Denissen & Miller, 2007; Montiglio, Ferrari & Réale, 2013).

There are several approaches to explaining the ultimate causes of human and animal personality variation (where animal personality is also called as coping style, behavioural syndrome, etc. (Wolf et al., 2007)), and though the available methods and background knowledge are quite different in the two areas, the basic explanations are rather similar. These include trade-offs between different life-history strategies (i.e., reproducing early vs. growing more) (Wolf et al., 2007) or personality dimensions (Nettle, 2006), individual niche specialisation (Bergmüller & Taborsky, 2010; Montiglio, Ferrari & Réale, 2013) and different forms of fluctuating or frequency dependent selection (Penke, Denissen & Miller, 2007).

However, the possibility that selection for cooperation can also play a role in causing or maintaining this diversity has not yet received proper attention. We think that there are many social situations with dividable tasks that require mutualistic cooperation; that is, cooperation from which all participants benefit, where behaviourally diverse groups must be better off. Examples for such dividable tasks could be the operation of hives, webs, etc. (Buech, 1995; Duarte et al., 2011; Grinsted et al., 2013), the rearing of offspring (Clutton-Brock, Russell & Sharpe, 2004; Barta et al., 2014), group hunting (Stander, 1992; Newton-Fisher, 2014) or even trade in human context (Gowdy & Krall, 2013).

We propose that division of labour between socially related conspecifics can cause and maintain variation in behaviour through negative frequency-dependent selection that continuously decreases a phenotype’s benefit as its frequency increases. Selection thus creates diversity, by favouring the rare, acting on preferences and skills; that is, on personality traits. The aim of this article is to study such a strategic model of division of labour. But first, because the theoretical and empirical literature on division of labour (henceforth DL) is so diverse, we find it useful to give a short outline and classification of the phenomena.

DL, which is a cooperative act where the acting parties are engaged in different tasks, can be observed on every level of organization from genes to social groups (Maynard Smith & Szathmáry, 1995). However, the literature is somewhat confusing in what DL actually is, as authors put the same label on very distinct phenomena or mechanisms. Therefore, we will categorize DL along two dimensions: whether it is linked to reproduction or is unlinked to reproduction, and whether it occurs within an individual or between individuals (Table 1.). Reproduction-linked DL means that it directly arises from some kind of reproductive DL. Sensu stricto reproductive DL means that some members of a group are reproducing, while others are not (Simpson, 2011) like the sterile castes in eusocial insect societies or the somatic cells after the germ-soma differentiation in multicellular organisms. We think that reproductive DL in a broader sense happens between other-sex individuals too. These could allow reproduction-linked DL to arise, such as the functional and physical specialisation of reproductively inactive female ants and Trematoda larvae to defence (Duarte et al., 2011; Lloyd & Poulin, 2014), and also any sexually breeding pair of parents taking mildly or strongly different roles in rearing their offspring (Buech, 1995; Barta et al., 2014; Gurven et al., 2009).

Table 1 A possible classification of the diverse phenomena labelled as division of labour (DL). DL can occur within or between individuals and can be linked to reproduction or unlinked, that is independent of it (for further details see the main text). This paper concentrates on the section with grey background that we call social DL.

	Reproduction-linked	Reproduction-unlinked	
Within ind.	– Germ-soma differentiation in Volvox colonies (Michod et al., 2006)	– Paralogous genes (Rueffler, Hermisson & Wagner, 2012)	
	– Non-reproductive polyps in Hydrozoa colonies (Cartwright, 2003)	– Enzymes in a protocell (Boza et al., 2014)	
		– Specialized limbs (Rueffler, Hermisson & Wagner, 2012)	
Between ind.	– Sterile workers in Hymenoptera societies (Sherman et al., 1995)	– Cooperative hunting in chimpanzees (Boesch, 2002)	
	– Sterile Trematoda rediae (Lloyd & Poulin, 2014)	– DL between reproductive female spiders in a common web (Grinsted et al., 2013; Wright, Holbrook & Pruitt, 2014)	
	– Task specialisation due to the reproductive hierarchy in meerkats (Clutton-Brock, Russell & Sharpe, 2004)	– Task specialization of sterile Hymenoptera workers (Duarte et al., 2011)	
	– Biparental care in different animal societies (Barta et al., 2014)		
	– Sexual DL in human societies (Gurven et al., 2009)		

Many examples of DL also occur between reproductively similar (or nearly identical) members of a group and therefore are not the direct result of reproductive asymmetries: we call this reproduction-unlinked DL. Examples include complementary enzymes in a protocell (Boza et al., 2014), the specialisation of limbs or organs (Rueffler, Hermisson & Wagner, 2012), task-specialisation in social spider colonies (Wright, Holbrook & Pruitt, 2014) and Hydrozoa colonies (Cartwright, 2003), and the cooperative hunting of lions and chimpanzees (Stander, 1992; Newton-Fisher, 2014).

As the abovementioned examples show, DL is not restricted to groups of individuals: any cooperatively linked pair or group of any kind of organisation (i.e., organelles, cells, tissues, etc.) can be involved. Hence, we differentiate between within and between individuals (Table 1). Naturally, boundaries between the categories outlined above and in Table 1 are sometimes blurred and controversial, yet we think that such a division still helps to see differences and similarities between these interconnected phenomena.

Each of the four categories circumscribes a type of DL that has been studied on its own (for examples and references see Table 1.), but one can also unfold a bigger picture with several interrelations between categories. For instance, the process by which between individuals DL becomes within individual DL is where a higher level or organisation emerges from a cooperative group of lower level organisms (or organizations). This mechanism often contributes to major transitions in evolution (Maynard Smith & Szathmáry, 1995). Classic examples are organelles constituting the eukaryota cell, the differentiation of germ and soma in multicellular animals and the appearance of sterile castes in eusocial insect societies, which are known as superorganisms (Maynard Smith & Szathmáry, 1995). Eusocial insect societies do not just represent a major evolutionary transition, but are also good examples for the blurred boundaries between the divisions outlined above, as they can be seen as one superorganism with an incomplete germ-soma differentiation and a flexible functional specialisation or as a social group of insects with strict reproductive DL and highly evolved DL among the non-reproductive members of the group with both physical and functional specialisation. Therefore, they can be put in both rows, that is in all elements of Table 1.

In this paper we are interested in reproduction-unlinked DL between individuals, which we will call social DL. We want to make a general argument about the possible role of social DL in creating and maintaining personality variation, as we find the literature lacking in this area. We present two variants of an analytically tractable model of DL. We show that skill differentiation is typical in both cases, even though they are not qualitatively nor quantitatively identical. Then we study an individual-based model representation of the problem to support and widen the analytical results, and finally we show how this mechanism can lead to diverse personalities or behavioural syndromes.

The Analytical Model

Model definition

Our model studies a large well-mixed population, where members of small groups solve a collective task. Solving this task produces a common good for the group which is distributed among the members after performing the task. Groups consist of a fixed number of (N) individuals sampled randomly from the population. Individuals can choose between two kinds of subtasks, A and B, which we will also call strategies. When entering a group, individuals make their subtask choice before they observe the choices of others, and task switching is not allowed. Individuals can specialize more or less to one strategy, but then they necessarily loose some of their flexibility: the more an individual specializes on one subtask, the less is his cost for performing it and more his cost for performing the other subtask. Hence, the abilities to perform the two strategies are not independent, and are in negative trade-off.

Individuals are characterised by two heritable traits: the probability of choosing A or B strategies, and the degree of specialisation to A or B. For simplicity, we assume that the population is asexual, so progenies own the same trait values as their parent did. We apply the methodology of adaptive dynamics to analyse the course of evolution. Our main interest is whether DL emerges as a result of an evolutionary branching into two subpopulations specialized for the two subtasks.

Costs

Each individual is characterised by the propensities of choosing each subtask and the costs assigned to performing them. Costs cA and cB, associated with tasks A and B, respectively, are constrained by the trade-off (1) cAα+cBα=kα

(Fig. 1), where k > 0 represents an additional cost to a basic minimal cost (which is scaled to be zero in the model). α > 1 means that the average cost is higher when individuals play mixed strategies (i.e., choose both subtasks with nonzero probability) than when they play only one (pure) strategy, A or B, with probability 1. The opposite is true when α < 1. We will choose cA as the independent variable, therefore (2) cB=kα−cAαα.

Observe that the derivative of this dependence at the two extreme points is either zero or infinite for α ≠ 1.

Figure 1 The characteristic trade-offs and cost-minimisation.

The thick lines are the trade-off curves for α = 0.5 (solid), α = 1.0 (dashed) and α = 1.5 (dotted). In general, trade-off curves are convex for α < 1 and concave for α > 1. Thin lines represent iso-c¯ lines of given q values (here q = 0.4). Higher the position of a line, the larger average cost, c¯, it means. As we are looking for the lowest c¯, we look for the lowest points where these iso-c¯ lines touch the trade-off curves at least in one point. For α < 1, we get the minimal c¯ where the iso-c¯ line is a tangent of the concave trade-off curve, the coordinates of the one common point define the optimal cA and cB. The maximal extremum we get where the iso-c¯ line is a tangent of the convex trade-off curve. For α ≥ 1, the minimum is one of the endpoints of the scale.

Benefits

The total benefit of a group, β(i), depends on the number of members (i) playing A (and consequently the number of members (N − i) playing B). β(i) is a function typically with one maximum in {0, 1, ..N}. That is, there exists a distribution of A and B tasks where solving the collective task brings the highest possible benefit. We assume that β is symmetrical, i.e., (3) βN−i=βi,

and has a single maximum in the middle.

We will study two types of models differing only in their way of benefit sharing. In model (I), the group benefit, β(i), is shared out equally among the group members, which could be a model of a cooperatively acquired common good from which nobody can be excluded. In model (II), β(i) is first split into two and then, within the subgroups playing A or B, halves of it are shared out equally, representing a sharing system, where the scarcer strategy gives a higher return. Therefore the individual benefits in the subgroups A and B are (4) bAi=βi/Nfor model (I)12βi/ifor model (II)

(5) bBi=βi/Nfor model (I)12βi/N−ifor model (II).

We discuss these two models together until differences emerge. In the following, we will dismiss the term 1/2 from the model (II) calculations, as it makes only a quantitative difference. Note that (6) Δbi=bAi+1−bBi

is the benefit advantage (or disadvantage) of playing strategy A over strategy B for an individual, provided that the rest of the group have i members playing A. It is easy to see that (7) Δbi=−ΔbN−1−i,

which we will use later on.

Payoffs

The two independent heritable traits of individuals are q ∈ [0, 1], the probability of choosing subtask A, and cA ∈ [0, k], the cost assigned to this task. The expected payoff values for an individual playing both strategies are: (8) wAq¯,cA=∑i=0N−1fi,N−1q¯bAi+1−cAwBq¯,cA=∑i=0N−1fi,N−1q¯bBi−kα−cAα1α,

where q¯ denotes the average q value of all the other population members, i.e., the probability of a randomly chosen individual playing A. The value for cB was substituted. The probability of having i individuals playing A among the other N − 1 group members is given by the binomial distribution (9) fi,N−1q¯=N−1iq¯i1−q¯N−1−i

obeying the symmetry relation (10) fi,N−1q¯=fN−i,N−11−q¯.

Finally, the expected payoff for an individual with trait values {q, cA} is (11) w¯q,cA,q¯=qwAq¯,cA+1−qwBq¯,cA.

Cost optimisation

The sub-problem of cost-minimisation of an individual with a fixed q value can be considered separately, because it depends only on the q value of the individual and contributes to the payoff additively. Therefore, our first interest is to find the optimal cA cost strategy that minimizes the average cost (12) c¯cA=qcA+1−qcB=qcA+1−qkα−cAα1α

for a fixed q.

For α = 1 it reduces to (13) c¯cA=qcA+1−qk−cA=k1−q+2q−1cA,

which is monotonous in cA. For q<12 it means that cost minimization results in cA = k (full specialisation for task B), while q>12 makes cA = 0 (full specialisation for task A) the optimal strategy. For q=12, cA is a neutral trait, which does not affect the average cost.

For α ≠ 1 we can look for an internal extremum determined by (14) dc¯cAdcA=q−1−qcAα−1kα−cAα1−α/α=0,

leading to (15) cAextrq=kq1−qα1−α+1−1/α

extreme cA value. This function decreases for α < 1 and increases for α > 1. Relatedly, as it can be seen in Fig. 1, this extremum is the cost minimum for α < 1 and the cost maximum for α > 1.

Hence, for α < 1, the cost optimum is internal and is determined by Eq. (15), while for α ≥ 1, it is at one of the end-points according to (16) cAminq=0,forq>12k,forq<12.

In the latter case the two end points are equally optimal for q=12.

The cost optimisation problem is trivial: q = 0, or 1, when the individual always choose the same subtask. Then it is optimal to fully specialize to the same subtask, leading to zero cost, and pessimal to specialize to the other subtask, leading to the maximal cost k. If q changes from 0 to 1, the optimal cA changes from k to 0. This transition is continuous for α < 1, but abrupt at q=12 for α > 1. It can be seen that the optimal c¯ increases with q until q=12 and decreases afterwards.

Singular points, invasion analysis

Now we turn our attention to the full evolutionary problem, the invasion of a mutant strategy and the emergence of DL, which is inherently frequency-dependent, as the benefit of an individual no longer depends only on the traits of its own, but on the traits of its opponents too. Therefore, we apply the methodology of adaptive dynamics (Geritz et al., 1997). In particular, we consider pairwise invasion, because it is known that continuous evolution (i.e., evolution via small muational steps) is controlled by invasion (Meszéna et al., 2005; Geritz et al., 2002; Geritz, 2005).

We describe an individual with the vector x: x = {q, cA} and assume that the population has two types of members, a common (resident) and a rare (mutant) type. Denoting the mutant traits by prime marks, the relative fitness of the mutant is (17) Wx′,x=w¯q′,cA′,q−w¯q,cA,q,

which is 0 by definition for x′ = x. (Since the mutant is rare, it does not modify q in the leading term.)

Note that the fitness is linear in q′, but is not so in q. Linearity in a rare trait is a common feature of game theoretic models with mixed strategies (Hofbauer & Sigmund, 1998), leading to non-standard behaviour relative to standard adaptive dynamics (Meszéna et al., 2001). On the other hand, the non-linearity in q is different from those commonly experienced in matrix games, and it is the consequence of the focal individual not playing against another individual, but instead, against a group of individuals.

Following the standard analysis we are interested in the singular points of the dynamics and their stabilities (Geritz et al., 1997; Leimar, 2009). x∗ is a singular point, if (18) ∂Wx′,x∂xi′x′=x=0,

i.e., if (19) ∂Wx′,x∂q′x′=x=wAq,cA−wBq,cA=0,

and if (20) ∂Wx′,x∂cA′x′=x=−dc¯cAdcA=0.

We have already solved Eq. (20) by Eq. (15) for α ≠ 1. Substituting it into (19), q is determined by (21) Gq+Hq=0

at the singular point, where (22) Gq=∑i=0N−1fi,N−1qΔbi

is the expected benefit advantage for the focal individual to play A, instead of B, and (23) Hq=−dc¯cAextrqdq=kq1−q11−α−1q1−qα1−α+1−1α

is the contribution from the cost term, valid for α ≠ 1.

It follows from the symmetry of Δb(i) and Δfi,N−1(q) (see (7) and (10)) that G(q) = − G(1 − q), so G(1/2) = 0 for model (I) and model (II) as well. One can see that H(q) is also zero for q = 1/2; that is (19) is satisfied for q = 1/2 at α ≠ 1.

The case α = 1 requires separate consideration, as the internal extremum of the average cost cA does not exist in this case. However, recall that cost is a neutral trait for α = 1, q=12. That is, the singular point is replaced by a singular line at q=12. This situation is usually neglected in adaptive dynamics.

Now we study the possibility of other singular points. It is easy to see that G(q) > 0 if q < 1/2, and consequently G(q) < 0 if q > 1/2. The same is true for function H(q) for α > 1, because c¯ has a minimum at q = 0. Therefore, the only solution of G(q) = 0 is q = 1/2. Further, similar to G(q), H(q) = − H(1 − q), and it can be shown that if α < 1 (α ≥ 1), then H(q) is strictly monotonously increasing (decreasing) with (24) limq→0Hq=−θk

(25) limq→1Hq=θk,

where (26) θ=−1,for α>10,for α=11,for α<1,

that is, for α = 1 H(q) ≡ 0. Consequently, q∗ = 1/2 is the only solution of (21) if α ≥ 1, so there is no other singular point except x∗. In contrast, for α < 1 other singular points are possible too. It is easy to see that because of the continuity and the symmetry of the functions G and H, if (27) G1+H1>0

(28) dGq+Hqdqq=1/2<0,

then at least one additional pair of singular points exists besides x∗ (Fig. 2). Denote these points by xu∗=qu∗,cu∗, where qu∗>1/2 and xu∗∗=qu∗∗,cu∗∗, where qu∗∗=1−qu∗<1/2, because of the symmetry of G and H. (If more than one such a point pair exists, then it is assumed that x∗ has the biggest and x∗∗ has the smallest equilibrium frequency.) Based on these results and assumptions, in the following subsection we investigate the invasion of a mixed strategy solving both subtasks in a resident population, where only one of the pure strategies is present as the resident strategy.

Figure 2 Function G(q) + H(q) on a schematic figure with singular points xu∗∗, xu and xu∗, when (Eq. (28)) are valid.

Invasion of a pure strategy by a mixed one

We consider the case when only one pure cost-free strategy is present initially; that is, the evolution begins either from the x(1) = {1, 0} or from the x(0) = {0, k} state, and we are interested in whether a mutant mixed strategy solving the alternative subtask with nonzero probability can invade or not. (Here, cost-free means that the subtask has no additional cost, since we can assume that any subtask has a constant basic cost which we can build in the benefit.) Without losing generality, let us look at the case of x(1), that is, every individual chooses subtask A and this subtask is cost-free. Now we study the invasion of a mutant in the neighborhood of x(1) by asking whether there are x′=q′,cA′ mutants with higher fitness than the resident one, that is whether (29) Wx′,x1≈∂Wx′,x∂q′x′=x1δq+∂Wx′,x∂cA′x′=x1δcA

can be positive, or not, where δq = q′ − q(1) = q′ − 1 ≤ 0 and δcA=cA′−cA1=cA′≥0 in such a way that at least one of these terms differs from zero. It is clear that the partial derivatives become (30) ∂Wx′,x∂q′x′=x1=G1+k,

(31) ∂Wx′,x∂cA′x′=x1=−1.

The second term in (29) is always negative (−1 × δcA < 0). Thus W(x′, x(1)) can be postitive only if G(1) + k is negative (since δq < 0), which is the condition for the invasion of a mutant (if δcA = 0 then q′ < 1 spreads). The sign of G(1) + k depends on the parameters and the type of the model. Depending on the model type we have two cases: (32) G1+k=βN−βN−1/N+k, for model (I),βN/N−βN−1+k,for model (II).

Considering model (I) it is clear that β(N) − β(N − 1) < 0, thus (32) can be negative. However, if N is large enough and/or the total cost k is high enough, then G(1) + k > 0 so W(x′, x(1)) < 0 and thus x(1) is resistant against the invasion of mixed strategies. The situation is a bit different for model (II). Again, β(N)/N − β(N − 1) is negative, but this value is smaller than in the previous case, thus G(1) + k can easily remain negative if N is large and β(N − 1) > k.

Because of the symmetry in the model, the condition for the invasion at x(0) is qualitatively the same as at the point x(1). Here if G(0) − k > 0, then a mixed strategy mutant can spread, which is more probable in model (II), as before. However, we must admit a mathematical nuance at this point. The trade-off function (2) has a zero-or-infinite derivative at cA = 0, or 1, for α ≠ 1. That is, if mutation steps are uniformly small, when parametrised by cA, they become non-uniform when re-parametrised by cB. From a biological point of view, the natural assumption is that mutations are uniformly small in cA in the vicinity of cA = 0 and uniformly small in cB in the vicinity of cB = 0. Thus cB, the cost of the mutant B strategy must be the basic variable at x(0).

Further progress after the initial invasion in the direction of decreasing q is very different for α < 1 and for α > 1. So let us assume that invasion is possible. Then a new mutant x′ spreads and fixates in the population. Clearly q′ < 1 and cA′≥0 in this new resident state. To study the invasion of the next mutant x″ we can follow the same analysis as before. However, the partial derivative of the fitness difference as a function of cA″ is (33) ∂Wx″,x′∂cA″x″=x′=−q′+1−q′OcA′α−1,forα>1,1−2q′,forα=1,−q′+1−q′/OcA′α−1,forα<1

(cf. (14)). If α < 1 then this derivative is a large positive quantity for very small cA. Therefore cA″>0 easily spreads, and the invasion process continues to the inner singular point. Recall that the cost optimum for a given q is internal for α < 1. Consequently, the “best” direction for a small mutation follows the direction of the curve cAextrq. That is, we have an essentially one-dimensional dynamics along this curve; direction is determined by the sign of −q′ ≈ − 1, thus invasion of cA″>0 is not supported and invasion events stop in xm where qm < qm−1, cAm=cAm−1=0. Because the cost optimum is always at one of the end points of the cA scale, if the pure strategy is invaded, then evolution proceeds along the edge cA = 0 of the strategy space. Direction of this one-dimensional strategy is determined by the sign of G(q) + k, as the cost strategy does not change; a rest point is reached when this variable becomes zero. As G12=0, the rest point xm is reached at q>12, i.e., before the cost optimum switches to the other extreme.

Convergence and branching of the singular point x∗

The question is the x∗=q∗,cA∗ singular point’s convergence stability and the conditions of branching. The convergence of a singular point means that in its neighborhood, evolution leads the system towards this singular point (Geritz et al., 1997). However, when close to it, branching gives rise to two different phenotypes in the evolutionary optimum. This is the state where coexistence of different types or DL is observed. Following Leimar (2005) and Leimar (2009), to study convergence we have to find out whether the Jacobian matrix J = H + Q of the selection gradient is positive or negative definite, where (34) Hi,j=∂2Wx′,x∂xi′∂xj′x′=x

(35) Qi,j=∂2Wx′,x∂xi′∂xjx′=x,

and xi denotes the coordinates of the x vector. If J is negative definite, then x is strongly convergence stable. That is, this state is convergence stable independently of the characteristics of covariance among the mutations in x1 and x2. Further, if H is positive definite or indefinite, then the singular point is a branching point (Leimar, 2009).

It is easy to see that H1,1 = Q1,2 = Q2,1 = Q2,2 = 0 in our model, consequently the J matrix becomes (36) Q1,1H1,2H2,1H2,2.

We have seen that if β(i) is symmetrical then q∗,cA∗=12,k2α becomes a singular point. Then the J matrix becomes: (37) Δwq∗−2−2α−1cA∗,

where Δw(q∗) is (38) Δwq∗=∂wAq,cA−wBq,cA∂qq=q∗,cA=cA∗.

After the derivation we get (39) Δwq∗=∑i=0N−1N−1iq∗i−11−q∗N−2−ii−N−1q∗Δbi.

Using q∗ = 1/2, the symmetry of β(i) and Eq. (22) we get (40) Δw1/2=12N−3∑i=0N−1N−1iiΔbi.

Since β(i) = β(N − i), it is easy to see that Δw(1/2) < 0 for both models.

The q∗,cA∗=12,k2α point is convergence stable (eigenvalues of (37) are negative) if (41) Δw1/2+α−1cA∗<0Δw1/2α−1cA∗−4>0.

Since Δw(1/2) < 0, the second criterion can be true only if α < 1. However, in this case, the first criterion will always be true. So the conditions for the singular point’s convergence stability will be (42) Δw1/221/αα−14>k.

We note here that Δw(1/2) is greater for model (I) than for model (II) at the same β(i), thus the above criterion can be satisfied with a smaller α (stronger trade-off) in model (I) than in model (II). Similarly, Δw(1/2) is greater if the maximal value of β(i) is smaller, that is, branching needs stronger trade-off if the benefit to cost ratio of the collective task is smaller.

We can also get the H matrix using the formerly shown calculations: (43) 0−2−2α−1cA∗,

It is easy to show that H is indefinite, therefore the singular point will always be a branching point. Obviously, the indefiniteness is related to the fact that the cost is minimal at the corners {1, 0} and {0, k} but maximal at {1, k} and {0, 0}.

To summarize: the q∗,cA∗=12,k2α singular point can only be convergence stable if α < 1, which is fulfilled if condition (42) is true. In this case the singular point is a branching point too. If α ≥ 1, both J and H will be indefinite. In this case, the convergence of the singular point depends on the covariance matrix of the mutants (Leimar, 2009).

Individual-based Simulations

Our individual-based numerical models were designed to be as similar to their analytical counterparts as possible; they differ only in being somewhat closer to natural systems. We present our methods and results using one arbitrary but illustrative parameter set concerning the size of the groups, the population and the benefit function, but we note here that the presented results are robust against variation of the parameters.

As in the analytic models, each individual is characterised by the strategy pair x = {q, cA}. In each round, N = 7 individuals were sampled randomly from the well-mixed population of 1000 individuals to face the collective task. In each generation, 1000 such groups were selected, so that individual fitnesses were updated on average 7 times in each generation. The group’s success in solving the task depended on the ratio of members choosing each subtask in a way that the maximal benefit was achieved when this ratio was closest to 1. The actual benefit of the group was defined by the function βi=30σ2πe−i−μ22σ2 with σ = 5, μ = 3.5 and i denoting the number of group members playing one strategy. Varying σ, the behaviour of the models changes in a not monotonous, but still only quantitative manner, therefore we could use an arbitrary value for this parameter. The benefit defined by the above function was shared between group members as defined in the analytical models (I) and (II). In each case, payoffs were added to the group members’ fitness, and at the end fitness was divided by the number of games an individual participated in. Those who did not participate continued to have zero fitness. After the 1000 fitness updates, 100 bouts of reproduction took place; we compared the fitnesses of two randomly selected individuals, and the offspring of the one with the higher fitness replaced the other. Every offspring inherited its parents’ traits, but with probability 0.1, mutations occured in both traits of the offspring. The mutant trait values followed a uniform distribution around the parents’ trait values, so they became q′ ∈ [q − ϵ, q + ϵ] and cA′∈cAα−ϵα,cAα+ϵα with ϵ = 0.005 (and with the obvious limitations that 0 ≤ q′ ≤ 1 and 0≤cA′≤k).

Branching

For testing the predictions of model (I) and (II) regarding the branching points, we designed simulations to map the parameter space. We ran 20 repeats for each parameter-set and looked for the frequency of evolutionary branching. At the beginning of a repeat the whole population was uniform with q0 = (1 − q0) = 0.5 and c0=kα−c0αα=k2α to make sure that the trait values were in the singular point x∗=12,k2α. In each simulation, we followed the population for a previously estimated time, enough for branching in most cases: 1.5 × 105 or 105 generations in model (I) and (II), respectively.

At the end of the simulations there were three types of results: (1) the population was nearly uniform, evolving to x(0) or x(1), (2) branching occurred, half of the population became specialized in subtask A, and the other half in subtask B, and (3) the population was still distributed around the original states, probably meaning that there was not enough time for branching.

After mapping a huge part of the parameter space we concluded that the simulated results (Fig. 3) match the three most important predictions of the analysis; furthermore, they reveal a new charachteristic of the stochastic model. First, when branching occured, it happened in or very close to the singular point (Fig. 4B). Second, the frequency of branching followed qualitatively the analytically calculated curves (42) in both models. Third, in model (I) specialisation occured in a much narrower area of the parameter space than in model (II), as predicted in (40). We also have a fourth observation, that is, in model (II), above the predicted border line, the closer α is to 1, i.e., the weaker the trade-off is, the more branching occurs, probably as a result of the strong negative frequency dependent selection. Furthermore, while the invasion analysis is valid only in the close neighborhood of the singular point, or close to the x(0), x(1) points, the numerical simulations show that local analysis successfully forecasts global behaviour. On the other hand, branching can occur above the critical α values as well, although with significantly smaller frequency than below the critical α (Fig. 3B), because the x∗ singular point remains a branching point for every α. Thus if the selection and the stochasticity of the system allows x to move into the vicinity of x∗, occasional branching can occur. Naturally this depends on the initial conditions as well.

Figure 3 Frequency of branching at different k and α values in model (I) (A) and (II) (B).

The solid curves are the predicted border lines, above which branching never occurs in the related analytical models. The darkness of the points denotes the number of branching events out of 20 at a given parameter combination with the darkest colour denoting 20 and the lightest denoting 0 branching events.

Finally, to see how much the benefit function counts in determining the branching behaviour, we designed some simulations with different asymmetrical benefit functions using parameter sets where branching was typical before. During these simulations we have seen that small asymmetries result in specialisation, as before, with subpopulation sizes resembling the amount of asymmetry, but large asymmetries result in homogeneous populations with only one pure strategy. The latter is probably a result of stochasticity and the singular point being too close to the unstable fixpoints x∗u or xu∗∗.

Initial values and convergence

After testing the existence of the predicted singular point x∗=12,k2α, we studied the effect of the initial population trait values, {q0, c0}, on branching in model (I). As we have shown in ‘Invasion of a pure strategy by a mixed one’, the x(0), x(1) end points can be stable against the invasion of mutants. Using the Gaussian benefit function defined above with not too small (>0.001) k values it is easy to see that [β(1) − β(0)]/N − k < 0 and [β(N − 1) − β(N)]/N + k > 0. Thus, there are additional singular points, xu∗ and xu∗∗, as assumed above. We pointed out that the x∗ = 1/2 singular point and the x(0), x(1) end points are convergence stable points, while xu∗ and xu∗∗ singular points are not. These additional singular points separate the phase-space and make the dynamics converging either to x∗ or to one of the two end points.

To test these predictions, we ran simulations from 11 × 11 different starting points using the formerly tested parameter set (k = 0.25, α = 0.15) that enabled branching. The results (Fig. 4) show us that the x(0), x(1) end points are indeed convergence stable such as the singular point x∗, and that there are unstable points separating the dynamics. Although our numerically estimated results do not exactly match the predicted xu∗=q=0.0154,cAα=0.168 and xu∗∗=q=0.984,cAα=0.644 points, they are quite close. We note here that in model (II) in cases when x(0), x(1) endpoints are unstable, there can be other singular points apart from x∗. These additional singular points do not modify our general result, that is the behaviour of x∗ and can be observed only at very specific parameters.

Figure 4 The effect of the initial trait values on branching in model (I).

Plot A. shows the intial parameter combinations with k = 0.25 and α = 0.15. Darker points denote events, where branching did not occur, and lighter points denote branching events. Plot B. shows two runs of g × 104 generations, those denoted by squares on plot A. with respective colours.

Discussion

We have highlighted how the social division of labour can lead to specialisation in a socially connected population. Our results not only tell us that such a mechanism can work, they also reveal the key contributing factors, such as the strength of the trade-off, the cost-benefit ratio, the type of the benefit sharing, and the difficulty of invading the state where every individual resolves to the same one of the two subtasks. Although most of these constraints of specialisation have been described by others (Wolf et al., 2007; Gavrilets, 2010; Goldsby et al., 2012; Rueffler, Hermisson & Wagner, 2012; Nakahashi & Feldman, 2014), we believe that the social division of labour is an important new context.

In this paper, we constructed a model framework for the emergence of consistent, heritable behavioural variation via social division of labour. We have shown analytically that if population members frequently face a collective task consisting of costly subtasks, where subtask efficiencies and the probability of choosing one of the subtasks are (genetically or culturally) heritable traits and group benefit is a function of subtask frequency, then a weak trade-off between subtask efficiencies (α < 1) can drive the population into specialisation. The specialised population members thus have optimal fitness, as DL enables them to get the highest benefit with the lowest cost, this way maintaining diversity. This occurs more easily if benefit sharing is based on the quality of contribution; that is, if those who represent a scarcer strategy, get a higher share (model II). We emphasize here that according to the general theoretical results (Leimar, 2009) branching is not impossible even if the trade-off function is concave (α > 1, trade-off is strong), but then branching depends on the covariance of mutations in q and c traits. We also verified our results and extended them by numerical simulations, where more realistic conditions, such as a finite population and stochastic events were met. We show that branching is possible in the individual based model even if the singular point x∗ is not convergent stable. We note here that although it would be biologically more appropriate to consider traits to be determined by many independent genes, the possibility of hybridization would complicate the outcome considerably. (There is no other difference between the two kinds of modelling, see (Taylor & Day, 1997)).

We have seen that one of the biggest hindrance of branching is the population being in the locally stable state of performing only one of the subtasks, that is, the invasion of the newly emerging strategy. Slightly different mutant mixed strategies cannot invade a population like this. However, in natural systems, mutations are not the only cause of behavioural change, since behaviour is shaped by environmental factors too. Our models do not include developmental plasticity, adaptations of a lifetime, environmental effects or learning but it is important not to forget about all these processes enabling behavioural shifts.

Our model describes two different types of social organisation differentiated by the way of benefit sharing. In social species, sharing a benefit collectively acquired is not rare: social spiders share a common web and their food captured together (Grinsted et al., 2013); several Hymenoptera species’ queens share the nest/hive founded together (Duarte et al., 2011), and lions or chimpanzees share their prey hunted together (Packer, Scheel & Pusey, 1990; Newton-Fisher, 2014). However, we have little knowledge of a sharing system where the members’ proportions are in relation to their specific value to the group. There is only one species where both empirical and experimental evidence shows that the sharing system often resembles that of model (II), and that species is us: humans (Leuven, Oosterbeek & Van Ophem, 2004; Kanngiesser & Warneken, 2012). Obviously, this is not the only reason why we find far more examples of social DL in human social systems than we find elsewhere, but it is still in concordance with the fact that DL is more easily performed. That is, both invasion of a mixed strategy and branching appeared more easily in model (II) than in model (I). Parallel to this observation we can also see that task specialisation without sexual DL is also much more frequent in human societies than elsewhere. Our hypothesis is that the system of an inherited, life-long task specialisation could easily contribute to the appearance of different behavioural syndromes, especially in humans.

In the literature there are several hypotheses about the emergence of personality and behavioural syndromes, in some aspects similar to our idea, and we are convinced that these process are not mutually exclusive. On the contrary, they are probably working together. One significant idea is social or individual niche specialisation (Bergmüller & Taborsky, 2010; Montiglio, Ferrari & Réale, 2013), which says that each individual occupies a niche, and the more diverse a social system is, the more and more specific social niches the society can maintain. At first sight, these two ideas, social niches and social DL, seem similar, but the basic situation is quite different: social niches are needed for weakening social conflict, i.e., conflict of interest, while social DL is needed for situations with mutual interest.

Our work has emphasised that not only conflicts, but cooperation can also lead to personality diversification. We find this idea to fit well with a history of human evolution where a positive feedback loop of social DL and behavioural and skill differentiation could have contributed to the current diversity of human personalities.

We thank Douglas Yu and the anonymous reviewers for their useful comments on the manuscript.

Additional Information and Declarations

Competing Interests

Author Contributions

The authors declare they have no competing interests.

Zsóka Vásárhelyi, Géza Meszéna and István Scheuring conceived and designed the experiments, performed the experiments, analyzed the data, contributed reagents/materials/analysis tools, wrote the paper, prepared figures and/or tables, reviewed drafts of the paper.

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
