# Peer review of "Evolution of heritable behavioural differences in a model of social division of labour"

_PeerJ, doi:10.7717/peerj.977_

## Round 0.1 · original submission · Minor Revisions

Thanks a lot for considering PeerJ as the source of publication of your best work. As you will see in the following section, some minor issues were raised by the reviewers that will requiere your attention. Once you consider them properly addressed, it will be my pleasure to review your improved manuscript.

Reviewer 1 ·

Basic reporting

1 the order of the authors on the first page differs from that in the second page. Please make it consistent.

Experimental design

No Comments

Validity of the findings

All the results are correct.

Additional comments

Vasarhelyi et. al. investigated how division of labor can pave the way for the diversity in traits. The model is somehow standard in the framework of adaptive dynamics. And the result is not unexpected.
When the trade off between sub tasks is strong enough, the evolution of coexistence of specialised individuals is likely to evolve.

1
I guess the result highly relies on the assumption on how to describe how strong the trade off is, Fig1.
It would be natural to assume that CA and CB are two independent variables, and investigate how this will alter the evolutionary divergence when they are both under the selection pressure. In the revised version, a short discussion along this line can benefit the readers.

2
Thought the authors are addressing how the emergence and maintenance of diversity is promoted by division of labor, I cannot see which part of the results refer to the emergence problem and which part refer to the maintenance problem. It should be clarified.

3 there are two traits, one is task-A-choosing probability while the other is the cost of CA. While I am sure the task-A-choosing probability should be a trait of an individual. it would be nice to explain why and how CA is a trait of AN INDIVIDUAL? In addition, it is necessary to explain why these two traits should evolve at the same time scale?

Reviewer 2 ·

Basic reporting

Overall, the manuscript presents the proper background, is easy to follow, well-structured and nice to read.

Some points that, in my opinion, can be improved:

- Line 119: why to study these two models of benefit sharing? Are they ubiquitous in nature? (even knowing details are provided after line 368, an intuition for these functions when they are defined would improve the reading of the manuscript).

- Also related with background concepts, I think that the paper would be enriched if more intuition behind the role of trade-offs were provided. A good job is done in the Discussion (lines 368-) where background related with the different benefit models is exposed. The authors could do the same regarding alfa, as they show that the possibility of branching strongly depends on this parameter. Some questions should be posed: What kinds of trade-off curves are more common in nature? There are previous works about the issue?

- Sometimes the authors refer to cooperation (e.g. lines 34, 47, 396). In the context of this work, I believe that the benefit function suggests a special kind of cooperation: mutualism, i.e. everyone in the group benefits from DL. A different kind of cooperation, altruistic, would suggest that some tasks have increased exogenous costs, imposed by the environment, and individual preferences through those tasks are not easy to explain. A note on that would be recommended.

- A great job is done in the categorization of different DL phenomenon. Yet, regarding clarity and unambiguity, I think that the choice of “unlinked” for a category is not the best one, as it does not translate, per se, any meaning. Reproduction-unlinked, or any other meaningful designation would be preferable.

- It would be beneficial for a broad range of readers if the authors provide a short definition of “negative frequency dependence”, given the central role of this concept in the work. Just a short note on line 42 would definitely fit.

Experimental design

The work is well motivated with the interesting conundrum of behavioral diversity and division of labor. The conclusions are strongly supported by both analytical and numerical evidence. The contents fit the scope of the journal (Biological Sciences).

Validity of the findings

The analytical demonstration and numerical validation are sound.

Some suggestions regarding minor points that could be explained:

- Line 287: why different generations for the different group benefit models? Is there any intuitive reasoning for the running times to be different?

- Line 268: the authors choose a continuous Gaussian function to model benefit. Lower deviations would generate benefit functions that would, relatively, give advantage to groups with the ratio of chosen subtasks close to 1. The advantage of groups that present both subtasks vanishes for high standard deviations. Do the authors have evidence that this parameter does not change the numerical conclusions? A note on this should be added.

- Figure 3: The authors demonstrate, analytically, that the singular point is only a branching point when alfa<1. Numerical results are only presented for alfa<=1, showing that indeed branching occurs. What happens, numerically, when alfa>1? Do the authors have the numerical results to enlarge Figure 3 B for (e.g.) 0.4 < alfa < 1.2, to show that branching for alfa>1 is indeed rare?

Additional comments

Please, find above the points that should be addressed, in order to improve the quality of the manuscript.

Some minor (possible) typos:

Line 96: reference to “both models”. What models? (different benefit sharing functions were not yet introduced, if was the case that the authors were referring to them)

Line 106: I imagine that the extreme points are cA=0 or cA=k. Yet, what are desirable limits of k? May k be negative?

Figure 1 caption: dashed line corresponds to alfa=1.

Equations 4 and 5: bA(i) and bB(i).

Line 52: reference to not linked, while unlinked is further used.

Lines 202 and 215: > appears after G(0)-k and cA’’, respectively. Is that a typo? If it is a specific notation, that should be introduced.

Equation 33: typo or unmentioned notation should be introduced.

---

## Round 0.2 · accepted · Accept

Congratulations for such an excellent work addressing all the issues raised by the reviewing process. Now your manuscript is ready for publication. I'm certain that this paper will attract much attention from PeerJ readers.